# Revisiting the Immunometabolic Basis for the Metabolic Syndrome from an Immunonutritional View

**DOI:** 10.3390/biomedicines12081825

**Published:** 2024-08-12

**Authors:** César Jeri Apaza, Juan Francisco Cerezo, Aurora García-Tejedor, Juan Antonio Giménez-Bastida, José Moisés Laparra-Llopis

**Affiliations:** 1Madrid Institute for Advanced Studies in Food (IMDEA Food), Carretera Cantoblanco 8, 28049 Madrid, Spain; 2Bioactivity and Nutritional Immunology Group (BIOINUT), Valencian International University (VIU), Pintor Sorolla 21, 46002 Valencia, Spain; 3Research Group on Quality, Safety and Bioactivity of Plant Foods, Campus de Espinardo, CEBAS-CSIC, P.O. Box 164, 30100 Murcia, Spain; jgbastida@cebas.csic.es

**Keywords:** metabolic syndrome, insulin resistance, macrophage, immunonutrition, obesity, type 2 diabetes

## Abstract

Metabolic syndrome (MetS) implies different conditions where insulin resistance constitutes a major hallmark of the disease. The disease incurs a high risk for the development of cardiovascular complications, and takes its toll in regard to the gut–liver axis (pancreas, primary liver and colorectal)-associated immunity. The modulation of immunometabolic responses by immunonutritional factors (IFs) has emerged as a key determinant of the gut–liver axis’ metabolic and immune health. IFs from plant seeds have shown in vitro and pre-clinical effectiveness primarily in dealing with various immunometabolic and inflammatory diseases. Only recently have immunonutritional studies established the engagement of innate intestinal immunity to effectively control immune alterations in inflamed livers preceding the major features of the MetS. However, integrative analyses and the demonstration of causality between IFs and specific gut–liver axis-associated immunometabolic imbalances for the MetS remain ill-defined in the field. Herein, a better understanding of the IFs with a significant role in the MetS, as well as within the dynamic interplay in the functional differentiation of innate immune key effectors (i.e., monocytes/macrophages), worsening or improving the disease, could be of crucial relevance. The development of an adequate intermediary phenotype of these cells can significantly contribute to maintaining the function of T_regs_ and innate lymphoid cells for the prevention and treatment of MetS and associated comorbidities.

## 1. Introduction

In recent years, non-communicable diseases (NCDs) have experienced a continuous increase, affecting an estimated 70% of all worldwide deaths in a year. This can be attributed to the dramatic increase in the prevalence of risk factors such as obesity, sedentary lifestyle, and altered food supply and preferences. The new epidemic in NCDs is related to the burden of metabolic syndrome (MetS), paralleling the worldwide increase in obesity, type 2 diabetes (T2D), cardiovascular diseases, cancers, and chronic pulmonary diseases. MetS is a complex disorder defined by a cluster of different interconnected factors, which incidence parallels that of immunometabolic diseases such as obesity and T2D. According to the World Health Organization [1], the global prevalence of obesity almost tripled between 1975 and 2016, when 39% of adults over 18 were overweight and 13% were obese. Similarly, the International Diabetes Federation (IDF) [2] estimated the global prevalence of T2D at 8.8% as of 2015 and is expected to increase to 10.4% in 2040. In this context, the IDF has established a worldwide prevalence of MetS at 20–25%. Also, the global prevalence of non-alcoholic fatty liver disease (NAFLD), affecting an estimated 24% of most populations [3], incurs a high risk for the development of T2D and of other major features of the metabolic syndrome (mainly cardiovascular complications like atherosclerosis and myocardial ischemia/infarction). Like obesity and T2D, M/NAFLD also showed a constant increase over the last 10–15 years, becoming the most common liver pathology worldwide. Community surveys reveal that the proportions of obese and NAFLD could even be aggravated when considering its distribution within specific ethnic groups [2,4]. The latter appears to indicate the relative importance that sociocultural factors can play on lifestyle, food supply, and preferences.

Insulin resistance (IR) constitutes a hallmark of the MetS and is recognized to be the core of the disease [5,6]. Notwithstanding, a carbohydrate–insulin model has been proposed to integrate multiple causal factors affecting (i) circulating fuels, (ii) hepatic fat accumulation, and (iii) intestinal function, leading the establishment of the diet-induced obesity [7]. Anyway, the key role of innate immune effector cells in determining the diet-induced obesity (i.e., innate lymphoid cells (group 2)—ILC2s) [8] and energy expenditure, as well as fat accumulation (i.e., monocyte-derived macrophages), has been revealed [9]. Taken together, these studies highlight the important role(s) of the intestine in the development of the MetS; however, there are no current strategies for MetS based on the selective functional differentiation of the innate effectors of the immune system. The intestinal mucosa supplies innate immune signals that influence the metabolic and immune processes responsible for the establishment of the IR and changes in the activity of potential beneficial environmental players, such as the gut microbiota. Clear examples are the critical signals underlying the innate immune ‘Toll-like’ receptor (TLR)-4 [10] and the role of innate and adaptive lymphocytes shaping lipid homeostasis and gut microbiota composition [11]. The so-called ‘gut–liver’ axis is sensitive to the gut microbiota, which affects energy supply and homeostasis, ectopic fat deposition, and intestinal and fatty tissue inflammation [12]. However, a causal relationship between the defined components of the gut microbiota and the disease has been largely inferential.

The bidirectional impact that MetS has on immunity and the key role that the immune system plays in the pathogenesis of different clinical conditions increasing the risk for MetS has been established [13,14]. Much of the interplay between innate immune system and hallmarks of MetS is attributed to the nucleotide-binding oligomerization domain-like receptors, or NOD-like receptors (NLRs) activation (i.e., NOD1) [15,16]. Both adipocyte cell models and in vivo studies on human adipose tissue have served to establish the association of insulin resistance to both NOD1 and NOD2 activities by eliciting pro-inflammatory signaling pathways [15,17]. These receptors contribute to determining the activity of recruited monocyte-derived macrophages in inflamed tissues, together with stimuli coming from the diet that are critical to promoting the intrahepatic fat accumulation and insulin resistance [9]. It has been reported that metabolically healthy obese subjects display significant differences at the activity of the innate branch of the immune system, mainly affecting the macrophage phenotypes [18,19]). In this context, abdominal fat represents an indisputable factor contributing to both over-boosted inflammation and the development of insulin resistance. It is worth bringing up here that macrophages’ activity could undergo epigenetic regulation on their inflammatory genes as well as TLR4 leading to immunometabolic adaptations. However, there is not any current food-based strategy targeting the activity of the monocytes-derived macrophages, which play dual roles, worsening or improving the disease, according to their sub-phenotype’s functional differentiation and activation. A recent review clearly states the potential role of nutrients to exert regulatory functions beyond its mere nutritional value through epigenetic changes [20].

Initiating events for the onset of MetS can stem even at early stages of the human development [21]. Overnutrition, even in the absence of pregnancy-related obesity, can promote metabolic dysregulation and predispose to T2D in the offspring [22]. There is evidence of the relevance of pre- and postnatal nutrition to development and interplay between the microbiota and the metabolic and immune systems that may result in an offspring phenotype susceptible to the metabolic syndrome [23,24]. Early “fetal or developmental programming of disease” through epigenetic changes is being increasingly thought as an integral underlying mechanism. It has also been reported that differences in the gut microbiota during the first year of life may precede the onset of metabolic alterations [25]. An inadequate gut microbiota composition and function in early life seems to account for the deviant programming of later immunity and overall health status. However, despite the relevant influence of gut microbiota on immune response(s), a review of the human clinical trials to evaluate effects of probiotics and synbiotics on obesity, insulin resistance, T2D, and NAFLD reveals the incomplete information on immune response(s) for the MetS prevention and/or treatment [26].

This review aims to provide a comprehensive picture of the diverse immunonutritional aspects affecting the monocytes-derived macrophages’ activity as a risk and consequence of the MetS. Particular attention is paid to molecular interactions as targets to prevent the immunometabolic alterations of the disease. An extensive search was conducted to identify as many studies as possible relevant to the role of innate immune effectors modulating either key metabolites or the activity of monocytes and monocytes-derived macrophages. Efforts were devoted to compiling human intervention studies as well as preclinical studies. All relevant studies are classified according to the description of specific molecular mechanisms affecting the activity of different components of the innate branch of the immune system.

## 2. Data and Methodologies

### 2.1. Data Source

An extensive search to identify as many studies as possible relevant to the literature review questions was conducted. In order to include all relevant information, multidisciplinary databases and information resources were explored. Unpublished research reports (“grey literature”) including dissertations or other scientific reports were not considered. The following information sources, databases, or search engines were used: Pubmed (biomedical), WorldWideScience (multidisciplinary), Web of Science (multidisciplinary), SpringerLink (multidisciplinary), Scopus (multidisciplinary), SciELO (multidisciplinary), and bioRxiv (preprint biology).

### 2.2. Methodology

The literature review process followed the following principles: (i) methodological rigor and coherence in the retrieval and selection of studies; (ii) transparency; and (iii) reproducibility.

To find a comprehensive set of scientific literature related to the immunonutritional strategies for the MetS on a target organism, either locally or systemically, different key words and phrases were used as search queries for each section of the tasks. Relevant scientific literature were to serve as the baseline information for the macrophage’s role in the MetS aimed for this procurement; their key elements were identified for those questions that needed to be answered in this scientific review.

The following general considerations were planned in order to reduce the risk of introducing bias into the literature review process and to assure the reproducibility of the review methodology. The search of different “key words” in different databases resulted in many duplicate documents. These duplicates were removed manually. For each subsection, different key questions were identified. The selection of relevant studies that might answer these identified questions was used to further refine the literature search performed when using the “key words” or phrases. Selected full-text studies were examined individually and manually for their eligibility to be incorporated in the Results section.

## 3. Prevalence and Genetic Risk Factors for the MetS

MetS is a complex disorder with a set of interconnected diverse risk factors, either environmental or endogenous (i.e., diet, microbiota, immunity and metabolic). This has favored confusion as to how to identify patients as well as define the prevalence and its variation within the different groups of a population [27,28]. Despite its current worldwide distribution, MetS appears more frequently in developed countries [29]. However, data vary depending on the source consulted.

Over the last decade, researchers have put intense efforts into determining the causes and prevalence of the metabolic (dysfunction)-associated fatty liver disease (MAFLD). It has become an increasing liver pathology worldwide, currently affecting an estimated 15–30% of most populations due to a dramatic increase in risk factors such as overnutrition, obesity, and sedentary lifestyle. The studies indicate an underestimated prevalence of the MAFLD and point to the need for including additional diagnostic parameters to the biochemical analyses from peripheral bloodstream. Non-inflammatory MAFLD incurs a high risk for the development of T2D and also takes its toll in regard to liver related morbidity and mortality. MAFLD constitutes the liver manifestation of metabolic syndrome. The prevalence of the MetS can vary by ethnicity, but most of the clinical studies (between 1980 and2015) highlight the absence of data concerning the influence of genetic and sociocultural factors on the disease’s prevalence [30]. Taking together with the genetic factors, others such as food supply and preferences, education, and access to health care can help explaining the MetS distribution within different ethnic groups. Otherwise, definition of the signals and immunometabolic events leading to the development of the severe consequences of the disease remains not fully described. It is worth it to bring up here that the relationship between MetS and MAFLD can be bidirectional; thus, MetS was identified as the major cause driving MAFLD in non-Hispanic black patients and Mexican Americans, but not for white Americans [31]. The influence of insulin resistance (i.e., HOMA index) on the risk of developing the severe form of the MAFLD has also been shown to display an ethnicity-dependent effect. For example, while Latino patients appear not influenced, the non-Latino, white patients are susceptible of suffering the disease [32].

Genetic makeup is related to the promotion or protection of MetS development and its related metabolic traits. An interesting study published in 2017 [33] described immune genes identified in genome-wide association studies (GWAS) related to metabolic phenotypes. Epidemiological studies have suggested that nucleotide variations in innate immunity genes could lead to type 2 diabetes mellitus (T2D) and associated metabolic disorders such as MetS [34]. The latter summarized an important number of studies describing gene polymorphisms coding for inflammatory mediators (produced by the innate immune cells) and their association with insulin action and the characteristics of the MetS [34]. Since then, an important number of human studies have gone deeper into the study of these polymorphisms and their relation to MetS (Table 1).

The “genetic/epigenetic–diet–microbiota” interaction is essential in the (not completely understood) pathogenesis of the MetS. One of the most important molecules involved in this interaction is the innate immune ‘Toll’-like receptor (TLR)-4, a cornerstone of the immune crosstalk between host cells and gut microbiota [10]. The genetic features of the host determine the intensity of the underlying molecular pathways and, therefore, the potential severity and pathological consequences of the disease. For example, the rs4986790 polymorphism of TLR-4 exhibits reduced receptor signaling and a lower inflammatory response, thus resulting in decreased atherosclerotic risk [48]. In addition to this, the prevalence of MetS, overweight-lipid syndrome and enlarged-waist-elevated triglyceride syndrome have also been related to this TLR4 gene polymorphism [47]. Otherwise, rs4986790 polymorphism appears to be associated with a reduced risk for T2D [49]. Recent research has also revealed the epigenetic regulation of TLR4 via the H3 lysine 4 trimethylation (H3K4me3) on the TLR4 promoter in diabetic macrophages [50] (Figure 1).

Previous studies have also associated the IL-6 polymorphism rs1800796 with obesity, as well as with components of the MetS (such as HDL cholesterol and Body Mass Index); although, its relationship with the disease itself remains elusive [37]. IL-18 is another cytokine related to the MetS [51], and its polymorphism rs5744292 is linked to impaired insulin sensitivity [42]. The anti-inflammatory IL-10, a cytokine biosynthesized in LPS-stimulated macrophages [52], and its polymorphisms (rs1800896; rs1800872; rs1800871) have also been related to MetS and diabetes [40]. Moreover, the pro-inflammatory TNF-α, one of the more investigated pro-inflammatory molecules regarding inflammation, has been shown to play a key role in the development of MetS. Thus, the polymorphism rs1800629 of the TNF-α gene is associated with higher plasma levels of this cytokine [53], supporting its relationship with hypertension and insulin resistance [35,36]. Anyway, different polymorphisms of the ADIPOQ gene appear related to the MetS in the Chinese population [43,46] (Table 1).

The challenge now is to increase our comprehension of how the polymorphisms of the innate immune genes determine the beneficial effects against metabolic syndrome and its related metabolic traits with diet. The question of whether MetS patients may obtain a benefit from the diet (i.e., Mediterranean diet) considering the genetic background has been addressed in a limited number of studies. In this regard, the evidence published indicates that the adherence to the Mediterranean diet reversed the gain weight linked to gene variants of the ADIPOQ gene in a group of volunteers [54]. The reduction in body fat in overweight volunteers (BMI ≥ 27) after the consumption of cloudy apple juice was related to its interaction with the rs1800795 of the IL-6 gene [55]. More recent studies have also shown a relation between Mediterranean dietary patterns and improved adiponectin and insulin levels, lipid profile, and C-reactive protein in obese patients (carriers of the rs3774261 and rs266729 ADIPOQ polymorphism) [56,57]. In another clinical trial [58], the interaction of the consumption of an olive-oil-enriched Mediterranean diet (compared with a low-fat diet) with two SNPs at the TNF-α gene promoter (rs1800629 and rs1799964) was described as a means to improve the markers related to metabolic syndrome. Notably, the consumption of the Mediterranean diet improved TG and hsCRP levels in rs1800629 patients compared to the group containing the rs1799964 polymorphism.

## 4. Innate Immunity’s Activity on the Insulin Signaling

Adipose tissue macrophages are key factors in obesity-induced insulin resistance by regulating a series of insulin-related and inflammatory-factor-related signaling pathways through paracrine interactions between adipocytes and macrophages. Prior research has demonstrated a role for type I interferons protecting against metabolic dysfunction and hepatic disease [59], inhibiting inflammasome activation [60], but its underlying molecular mechanism remains unclear, and its role in the development of IR remains unclear. For example, different interferon regulatory factors control adipogenesis processes, with IRF4 being a critical factor in lipid management by the adipocyte. IRF4 exerts regulatory actions on the inflammatory profile of macrophages, and its deficiency impairs insulin signaling [61]. Both interferon (IFN)-α and IFN-γ are known to exert ketogenic effects due to their stimulatory effect on lipolysis, despite showing dose-dependent effects on the hepatic fatty acid synthesis [62]. Recently, the effects of these IFNs have been extended to the promotion of the adipose tissue inflammation due to the fact that they reduce the infiltration of regulatory T cells into tissue. Inhibition of actively IFN-α-producing plasmacytoid dendritic cells appears as a key determinant to improve the metabolic indices and insulin sensibility by restoring the accumulation of PPARγ^+^ Tregs [63]. PPARγ, among others, constitutes a transcriptional factor promoting the macrophage polarization towards its M2 phenotype and displays an AMPK-signaling-dependent crosstalk with TLR4, contributing to anti-inflammatory effects [64].

In this context, macrophage bioenergetics and their mitochondrial function seem to play important roles on the macrophage’s metabolic rewiring. The macrophages’ bioenergetics undergo a switch towards an increased glycolysis (M1-like) and the citrate-derived itaconate, which is associated with lipid droplet accumulation. These response(s) are promoted by the acute iron deprivation, which is associated with inflammatory stimuli such as the hepatic production of bioactive hepcidin (*hamp*), favoring the immunoregulatory role of the endogenous itaconic acid. Using knockout models for the Irg-1 gene, it was demonstrated that by decreasing the production of this metabolite, fat accumulation is promoted [65]. This effect was associated with decreases in the fatty-acid-dependent phosphorylation. Thus, different strategies using the membrane-permeable dimethyl–itaconate could ameliorate the severity of palmitate-induced insulin resistance in C2C12 myocytes [66]. However, dimethyl–itaconate does not appear to be metabolized to itaconate intracellularly.

Previous research described a transition on the distinct populations of adipose tissue macrophages, during the development of obesity [67]. The predominant population exhibits a highly activated bioenergetic and displays a F4/80^hi^CX3CR1^+^ phenotype. Itaconic acid accumulates into F4/80^+^ macrophages [65]. CX3CR1 is predominantly expressed by F4/80^+^ macrophages but can also be found in hepatic stellate cells or endothelial cells and adipose-derived stem cells [68,69]. Serbulea et al. did not consider CX3CR1 as a significant contributor to the development of obesity. However, impairment of the CX3CL1–CX3CR1 axis has been reported as a significant promoter of inflammation and insulin resistance in male mice [70]. This contribution to the inflammatory processes appears to be associated with increased proportions of M1-like macrophages into adipose tissue. A similar effect was observed to be caused by the deficiency of CX3CR1 signaling in the macrophage migration and functional differentiation, aggravating the lipotoxicity in non-alcoholic steatohepatitis (NASH) mice [71]. Otherwise, recent research supports positive effects of CX3CR1^hi^ macrophages facilitating the survival and activity of the adipose-derived stem cells [68]. In this line, nutritional interventional preclinical assays showed that increased hepatic F4/80^+^CX3CR1^+^CD74^+^ macrophages improved the myeloid-derived antitumoral responses in mice under an HFD [72]. The latter authors could associate these effects to the administration of serine-type protease inhibitors (SETIs), which showed biological activity associated with their interaction with TLR4 (TRIF-dependent), modifying the hepatic profile of major lipids in hepatocarcinoma-developing mice (Rag2^−^^/−^ and Rag2^−^^/−^IL2^−^^/−^) [72]. Administration of SETIs to Rag2^−^^/−^ and Rag2^−^^/−^IL2^−^^/−^ under an HFD favored the reduction in hepatic triglycerides and glycemia. Anyway, in vitro assays revealed that macrophage bioenergetics were significantly modified to a less inflammatory profile when challenged to SETIs (Figure 2).

The aforementioned studies show that, when appropriately activated, macrophages can mediate positive and bidirectional interactions in the metabolic stress and its consequences. In this context, the capacity of TLR4, among others, as a sensor and modulator of metabolic stress, while at same time being affected by it, represents one of the greatest determinants of MetS and its consequences. However, and despite the intense research carried out on this receptor, there is little information about the immunometabolic signals that drive an adequate contribution between selective functional polarization of macrophages and their activity in the progression of MetS. For example, TLR4 plays critical roles in the polarization of macrophages toward their F4/80^+^CX3CR1^+^ phenotype and the production of interferons [73], which appears to occur in a delayed wave of signaling. On the contrary, its stimulation with its protype agonist gives rise to an acute and intense response, driving the underlying molecular signaling towards aggressive inflammatory responses, as well as the uncoupling of the immunological response(s) [74] (Figure 2).

## 5. Factors Determining the Innate Immune Adaptations in the MetS: Macrophage Polarization under Insulin Signaling

Dysfunctional immune response(s) in obese, diabetic, and NAFLD patients are quite well-identified [75,76,77]. In this context, metabolic disturbances on insulin signaling and its consequences in the glycometabolism occurring in MetS represent critical events to innate immunity activation, determining the typical chronic low-grade inflammatory state. Thus, conditioning the complex interactions that take place between tissues and the immune cells. As a consequence, aberrant immune response(s) aggravate the metabolic disturbances and disease severity. The enormous impact of immune disturbances on the main features of MetS, such as cardiovascular complications, atherosclerosis, and myocardial ischemia/infarction, has been identified [13,14].

The possible mechanism(s) underlying the link between metabolic imbalances and immune disturbances in MetS are diverse but mainly refer to alterations in nutrient supply, modulation of low grade inflammation [78], the intensity of the innate immune response affecting insulin resistance and impairing energy homeostasis, or of the adaptive immune response by binding to leukocytes receptors suppressing lymphocyte proliferation, to inhibit cytokine production of Th1-lymphocytes and to induce T-lymphocyte apoptosis. Moreover, monocyte/macrophage’s activity is tightly associated with a complex array of energy-demanding processes. These immune alterations can imply different organs were stem innate and metabolic signals representing potential origins leading to the loss immune cellular functions or aberrant expansion of the inflammatory processes. For example, macrophages have been identified as determinant cellular targets for the diet-regulated production of the platelet-derived growth factor (PDGF), controlling lipid accumulation in the adipose tissue and liver, as well as affecting insulin resistance [9]. To this end, knockout preclinical models have revealed that lacking the intestinal epithelial TLR4 receptor mice appear more susceptible to developing the MetS [10]. The latter authors suggested the regulation between the gut microbiota and intestinal cells affecting insulin production as the most probable cause. Anyway, MetS is associated with the impairment of uptake and levels of essential micronutrients such as iron [79], the availability of which has a profound effect on the anti-inflammatory (re)programming of macrophages [80]. Under iron deprivation, LPS (MyD88-dependent) macrophages polarization seems to be restrained, as well as their pro-inflammatory phenotype. Hepcidin plays a key role in iron homeostasis, limiting its intestinal absorption, as well as the availability of the micronutrient at organismal level, and is long known to be linked with insulin-resistant states [81]. Notably, hepcidin expression is upregulated, among other factors, by intestinal TLR4 activation [82]. Our preclinical studies on the hepatocarcinoma showed that immunonutritional agonists (i.e., serine-type protease inhibitors—SETIs) enable the upregulation of hepcidin production and decrease in ferritin concentrations and appear to be associated with elevated proportions of an M1-like macrophage phenotype, thus ameliorating tissue and metabolic impairment [83]. Insulin levels appeared slightly, but significantly increased without changes in the concentrations of peripheral glucose and NADH [83,84]. The underlying molecular basis of these effects could be suggested to be derived from the engagement of a TRIF/TICAM molecular pathway and the production of interferons.

Pro-inflammatory conditions developed in patients suffering MetS are known to favor environmental conditions leading to dysbiosis [85], thereby impacting the host metabolome [86,87]. These microbial imbalances contribute to the expansion, polarization, and activity of the immune effectors—monocyte-derived macrophages and innate lymphoid cells—towards specific sub-phenotypes, which reciprocally determine the composition of the microbiota [11,88]. Recent studies demonstrated that obesity-related insulin resistance is reduced in Rag2^−^^/−^IL2^−^^/−^ mice lacking innate lymphoid cells (ILCs) [8]. Meanwhile, ILC2s from intestinal origin in ILCs-carrying Rag2^−^^/−^ mice appeared as determinants of diet-induced obesity. Interestingly, small intestinal-derived ILC2s could be associated with a down-regulated expression of the angiopoietin-likeprotein4 (Angptl4) (Fiaf factor) [8] and tightly subjected to a microbiota-dependent regulation [89]. One mechanism by which the crosstalk between gut microbiota and ILCs can influence the development of the metabolic syndrome is the accumulation and/or mobilization of triglycerides. Taken together, innate immune imbalances can be linked to gut dysbiosis and metabolic dysfunction in the metabolic syndrome. 

A major consequence of the metabolic imbalances derived from the MetS is the promotion of cancer; therefore, it is possible that metabolic-influenced immune dysfunction significantly contributes to carcinogenesis. Numerous pathways that result in deleterious effects on the immune-mediated cancer editing and its elimination can be worsened by facilitating the metabolic survival of cancer cells and the lipogenic processes [90]. Macrophage-derived factors contribute to cancer resistance, partly by impinging on metabolically active tissues. Recent research using the Rag2^−^^/−^ and Rag2^−^^/−^IL2^−^^/−^ mice under hepatic pre-carcinogenic conditions showed that ILCs appear to play a significant role in influencing insulin production as well as macrophage polarization. Here, the administration of SETIs does not impair insulin production, as confirmed by the absence of TNFα production, but favors the concomitant polarization of monocyte-derived macrophages and ILC3 to ameliorate liver injury. It is worth mentioning here that low-grade inflammation in MetS lays at the cross-section of metabolic and inflammatory etiologies. In the liver and adipose tissue, recruited pro-inflammatory macrophages are identified as a major cause for the onset of insulin resistance [91,92]. Tissue-resident and recruited monocyte-derived macrophages are highly sensitive to microenvironmental metabolic changes in the liver and adipose tissue, which dictate transcriptional adaptations, determining the macrophage’s activity.

Synergizing with intestinal TLR4, the nucleotide–oligomerization domain receptor (NOD)-1 has also been revealed as a key partner in developing insulin resistance [93]. Moreover, another member of the NOD-like receptor family, the NOD, leucine-rich-containing family, pyrin domain-containing-3 (Nlrp3), has also been identified as a stirrer-up of obesity-induced insulin resistance [94]. Insulin may contribute to immunomodulatory events by regulating the activation of the NLRP3 inflammasome in monocyte-derived macrophages [95], but there is also a direct link between insulin resistance and increased NLRP3 expression [96]. These effects would be mediated by the production of IL-18 and interferon-γ. It is widespread, common knowledge that the activity of M1 macrophages produce interferons and the pro-inflammatory TNFα and IL-6 [97,98], which are closely associated with the progression and severity of insulin resistance. The role of recruited macrophages in tissue response(s) to insulin signaling is enormously dependent on their phenotype, as signaling pathways underlying cell surface receptors recognizing DAMPs or PAMPs are altered in pathogenic conditions such as diabetes [99]. Plasticity in the functional polarization of macrophages is reflected in the expression of different surface molecules, as well as transcription factors and the acquisition of epigenetic regulation based on environmental stimuli [100]. Thus, considering the critical role of intestinal TLR4 in the prevention of MetS seems to suggest that the functionality and role of macrophages could be influenced in the intestinal tissue. This could influence and/or “(re)program’m” a phenotype of these that facilitates or contributes to reducing the consequences and severity of insulin resistance in organs such as the liver, adipose tissue, and pancreas. In this line, the increased M1-like hepatic macrophage proportion without the production of these cytokines in Rag2^−^^/−^ and Rag2^−^^/−^IL2^−^^/−^ mice fed with SETIs [83] may interpret the data as showing protective effects of liver function. The latter effects could be triggered by signals stemming at intestinal level within the ‘gut–liver’ axis.

Taken together, the establishment of insulin resistance in the hepatic tissue also results in pathological levels of hypertriglyceridemia as well as an excess lipid accumulation and increased total and LDL cholesterol in patients suffering from the MetS [101]. In this context, macrophages develop a pro-inflammation due to the boosting effect of the MyD88-dependent pathway on the sterol-regulatory element-binding protein (SREBP)-2 activity [102]. As a consequence, the biosynthesis and import of cholesterol into macrophages are increased. Equally important to establish a pro-inflammatory response is the key role played by cholesterol homeostasis in controlling adequate processes for its resolution. The production of interferons either through the TRIF/TICAM signaling pathway and the NOD2 receptor can play key roles in conditioning cholesterol metabolism in the macrophages, and thereby their inflammatory response(s) [53,103]. As macrophages are close to the intestinal mucosa, their contribution to the expansion of ILC2s is plausible [104]. It is worth mention here that ILC2s serve as key determinants of diet-induced obesity [8] and can positively influence insulin resistance in the adipose tissue, thus limiting the severity of MetS. Intestinal ILC2s have also been identified as NOD2-sensing cells to selectively activate the inflammatory cytokine production [105]. Elegant reviews on the role of probiotics and synbiotics as co-adjuvants for the prevention and treatment of the insulin resistance syndrome can be found [26]; however, its benefits appear to be only partially supported by the data. At the liver level, recent pre-clinical studies showed the positive effects of the administration of SETIs on the hepatic accumulation of triglycerides and the major lipids profile (^1^H-RMN) of Rag2^−^^/−^ and Rag2^−^^/−^IL2^−^^/−^ mice fed a high-fat diet [72]. These effects could be associated with increased proportions of M1-like macrophages (CD68^+^F4/80^+^) and precursors of the ILC2s (CD117^+^KLRG1^+^) cells. Notably, the aforementioned interactions can be present as initiating events at all ages even in the earliest stages of life [21].

## 6. Concluding Remarks and Future Directions

MetS represents a global epidemy affecting all groups of the population either in developed or developing countries. It is a complex pathophysiologic state, thus requiring not a single therapeutic approach but several potential mechanisms contributing to its pathogenesis, affecting both metabolic and immune signaling. Today, it is recognized that treatment of MetS in childhood should focus more on preventive interventions rather than diagnosis of the disease. Understanding the genetic risk and environmental factors determining the sub-phenotypes of monocytes-derived macrophages could contribute significantly to developing strategies and implementing multidisciplinary public health policies to manage the MetS. 

Immunonutritional ingredients can significantly contribute to adequately maintaining the immunometabolic activities within the gut–liver axis and establishing an adequate tissue metabolic homeostasis. Thus, immunonutritionally based strategies can preserve the innate immune response(s) and potential imbalances of these, in the pathogenesis of different conditions like obesity and T2D, as well as MAFLD. Delineation of the role that immunonutritional ingredients can play within precision-nutrition-based strategies and others that are currently being studied may help to clarify the exact pathogenesis of the syndrome and expand the clinical criteria of MetS. However, integrative analyses and the demonstration of causality between IFs and specific gut–liver-axis-associated immunometabolic imbalances for the MetS remain ill-defined in the field.

Recent advances highlight a pivotal role for intestinal innate immune signaling to develop ‘trained immunity’, a de facto innate immune memory, which can help to balance tissue homeostasis and inflammatory processes associated with obesity, T2D and NAFLD. Moreover, the scientific evidence supports the critical role that environmental (i.e., diet, microbiota) exposures could play in immunometabolic health. Notably, the biomedical basis and underlying molecular mechanisms of these interactions remain ill defined. Characterization of the molecular events triggered by immunonutritional ingredients and thereby favoring a selective functional differentiation of innate immune effectors might be a future directive for preventing MetS. The possibility of developing ‘trained immunity’ influencing key immunometabolic events linked to MetS could provide the opportunity to prevent as well as repair alterations, to control the onset of the disease.

## Figures and Tables

**Figure 1 biomedicines-12-01825-f001:**
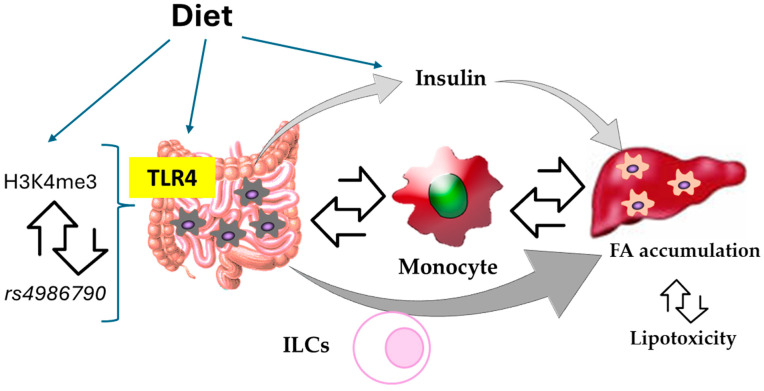
Schematic diagram for the interaction of the genetic and environmental factors influencing the control of hepatic fat accumulation by macrophages. ILCs, innate lymphoid cells; FA, fatty acids.

**Figure 2 biomedicines-12-01825-f002:**
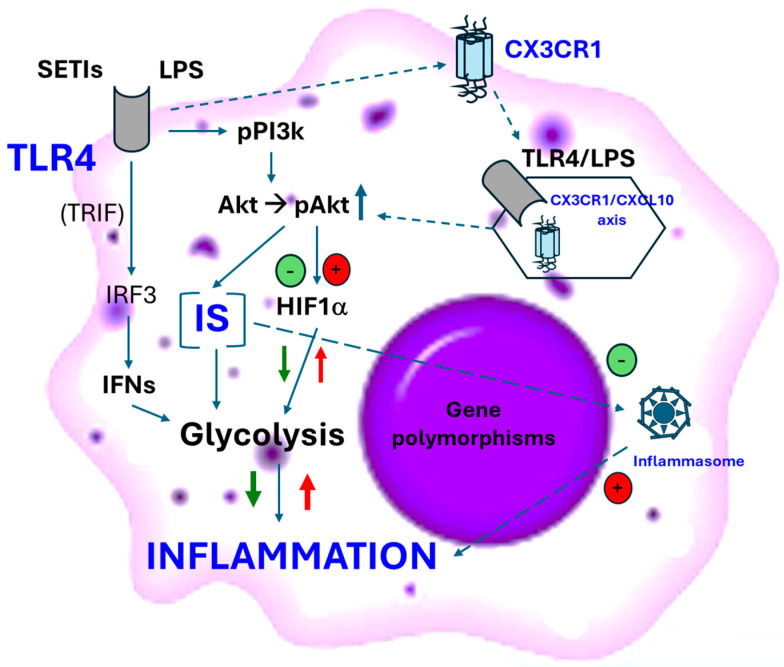
Schematic hypothesis to explain the potential different contribution of TLR4 to the inflammatory phenotype of macrophages in the metabolic syndrome. SETIs, serine-type protease inhibitors; LPS, bacterial lipopolysaccharide; IRF3, interferon regulatory factor 3; IFNs, interferons; IS, insulin sensitivity. Small-dotted line represents the sequence of events triggered by the activation of TLR4 by the prototypical agonist LPS (red symbols and arrows). Solid line represents the sequence of events triggered by the activation of TLR4 by SETIs (green symbols and arrows). Large-dotted line represents the influence of insulin on the NRLP3 expression.

**Table 1 biomedicines-12-01825-t001:** Gene polymorphisms of the molecules involved in the regulation of the inflammatory response associated with the metabolic syndrome.

Gene	SNP	Related Disease	Reference
*TNF-α*	*rs1800629*	MS, hypertension, insulin resistance	[35,36]
*IL-6*	*rs1800795*; *rs1800796*	MS, diabetes, obesity	[37,38,39]
*IL-10*	*rs1800896*; *rs1800872*; *rs1800871*	MS and diabetes	[40]
*IL-18*	*rs1946518*; *rs5744292*	MS and inflammation	[41,42]
*ADIPOQ*	*rs2241766*; *rs266729 rs3774261*; *rs6773957* *rs1501299*	MS	[43,44,45,46]
*TLR-4*	*rs4986790*	MS; insulin resistance	[47]

Abbreviations: ADIPOQ, adiponectin; IL, interleukin; TLR-4, Toll-like receptor-4; TNF-α, tumor necrosis factor-α.

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
