# Peer review of "Revisiting the Immunometabolic Basis for the Metabolic Syndrome from an Immunonutritional View"

_biomedicines, 2024, doi:10.3390/biomedicines12081825_

Round 1

Reviewer 1 Report

Comments and Suggestions for Authors

1. The manuscript reveals 385 grammar errors with 4% plagiarism.

2. Abstract: ", a better understanding of the IFs with a significant role in the MetS, as well as within the dynamic interplay in the functional differentiation of innate immune key effectors (i.e., monocytes/macrophages), worsening or improving the disease, could be of crucial relevance." The issue here is how does diet influencing the microbiota affect the immune system? Although many studies have demonstrated the positive impact of pursuing healthy behaviors on reducing disease risks, the combined effects of both dietary and physical activity components that drive the biological mechanisms at the molecular level through epigenetic modifications have not been widely studied.

3. What is the impact of diets on metabolic diseases through epigenetic modifications? For example, In a study conducted in a remote Swedish town, restrictive access to nutritional foods in one generation was associated
with mortality rates two generations later in a gender‐specific manner, i.e., paternal grandfather’s dietary supply was related to grandsons’ altered metabolism, and paternal grandmother’s dietary supply was linked to granddaughters’ altered metabolism. Another study suggested that the grand offspring of males, who were exposed to famine while in utero, suffered from higher BMIs than the control population.

4. What are the dietary compounds that affect different epigenetic DNA methylation events (see Georgel PT and Georgel P (2021).

5. Metabolism involving one‐carbon dietary constituents such as folate (B9), B vitamins (B2, B6, and B12), methionine and choline (betaine) are essential substrates or cofactors [https://doi.org/10.1093/jn/133.3.941S.]. These authors reported that inadequacies of these dietary constituents, contributing to classical dietary deficiency syndromes, can also cause neural tube defects, cardiovascular disease and cancer (https://doi.org/10.1146/annurevanimal‐
020518‐115206).

6. TLRs: Perkins et al. [https://doi.org/10.1089/jir.2016.0003] reported a review of the status of the chromatin profile expressions of inflammatory genes in response to TLR stimulation and how this chromatin landscape affects cell type specific and temporal gene expression. Notably, TLR can induce epigenetic
changes constituting both the positive and negative regulation of TLR‐induced
genes. However, signaling pathways and transcription factors play a pivotal role in chromatin‐based mechanisms affecting proinflammatory responses and epigenetic marks in rewiring context‐specific gene expressions in innate immune cell types. Please expand.

7. MetSyn: Please expand on the combined effects of diet and exercises on metabolic diseases through epigenetic modifications. What is the molecular mechanism involved?

Comments on the Quality of English Language

there are 385 grammar errors with 4% plagiarism.

Author Response

Dear reviewer,

We really appreciate your efforts reviewing this study. In accordance with your comments, we enclose our answers. The pertinent changes to the manuscript have been made, and a list of the modifications is enclosed.

  1. The manuscript reveals 385 grammar errors with 4% plagiarism.

Many thanks for your comment. The document has been revised by one of our colleagues to improve the text. Concerning the plagiarism, please, could you be more concise? Every idea from other authors is adequately quoted in the manuscript, in our opinion this is not plagiarism. Do you refer to the use of some sentences (English grammar) used by others? This could be a ‘false positive’ and cannot be considered plagiarism, just the use of a sentence that the authors used in their study and, as previously said, properly quoted in the manuscript. We never attribute to ourself any idea or conclusion previously stated by others.

  1. Abstract: ", a better understanding of the IFs with a significant role in the MetS, as well as within the dynamic interplay in the functional differentiation of innate immune key effectors (i.e., monocytes/macrophages), worsening or improving the disease, could be of crucial relevance."

The issue here is how does diet influencing the microbiota affect the immune system? Although many studies have demonstrated the positive impact of pursuing healthy behaviors on reducing disease risks, the combined effects of both dietary and physical activity components that drive the biological mechanisms at the molecular level through epigenetic modifications have not been widely studied.

Your comment is highly appreciated. We feel sorry for the misunderstanding. Concerning the microbiota, we fully agree about their ‘relative’ efficacy to drive selective functional differentiation processes of a specific population of the immune system. As stated in the manuscript the causal association of the gut microbiota to the disease is inferential (line 62-64).

As already stated in the manuscript (line 52-56) ‘…it has been revealed the key role of innate immune effector cells determining the diet-induced obesity (i.e., innate lymphoid cells (group 2) – ILC2s) [8] and energy expenditure as well as fat accumulation (i.e., monocyte-derived macrophages) [9]. Taken together, these studies highlight the important role(s) of the intestine in the development of the MetS.’ This is the reason why the document focuses on the description of molecular interactions and effects of the innate immune system in the metabolic syndrome.

In view of the reviewer’s comment the manuscript’s objective has been rewritten (line 95-98)

  1. What is the impact of diets on metabolic diseases through epigenetic modifications? For example, In a study conducted in a remote Swedish town, restrictive access to nutritional foods in one generation was associated with mortality rates two generations later in a gender‐specific manner, i.e., paternal grandfather’s dietary supply was related to grandsons’ altered metabolism, and paternal grandmother’s dietary supply was linked to granddaughters’ altered metabolism. Another study suggested that the grand offspring of males, who were exposed to famine while in utero, suffered from higher BMIs than the control population.

We completely agree with the reviewer’s comment. This is a very interesting and hot area of research, but unfortunately, this study did not focus on the epigenetic response(s) to immunonutrients. Within metabolic research, these are often studied in particular cell types or tissues, such as adipocytes, macrophages, skeletal muscles, or liver tissue, to name a few. It is known that histone deacetylase 4 is upregulated in macrophages by leptin and in skeletal muscle by interferon-γ, leading to a reduction in expression of inflammatory and energy expenditure genes, respectively (Cell Metab 19: 1058–1065, 2014). TLR4 is also known to be subject to epigenetic regulation, thus from a functional point of view, we could expect some epigenetic changes as macrophages acquire different phenotypes when exposed to the SETIs (line 216-228).

Taking into consideration the reviewer’s comment, some sentences have been included in the manuscript in relation to epigenetic aspects (line 79-85).

  1. What are the dietary compounds that affect different epigenetic DNA methylation events (see Georgel PT and Georgel P (2021).

According to the reviewer’s comment a brief sentence referring to these aspects has been included in the manuscript (79-85).

  1. Metabolism involving one‐carbon dietary constituents such as folate (B9), B vitamins (B2, B6, and B12), methionine and choline (betaine) are essential substrates or cofactors [https://doi.org/10.1093/jn/133.3.941S.]. These authors reported that inadequacies of these dietary constituents, contributing to classical dietary deficiency syndromes, can also cause neural tube defects, cardiovascular disease and cancer (https://doi.org/10.1146/annurevanimal‐020518‐115206).

It is known that macrophages’ subphenotypes are directly linked to their metabolism (Front. Immunol. 2014, 5, 420). Our intention focusing on immunonutritional factors is to state these displaying a ‘concrete’ mechanism of action or triggering specific signals, which could be used in the prevention/treatment of the disease beyond its mere nutritional value. For sure, the nutritional and lifestyle approaches of the metabolic syndrome are essential to control the severity and treat the disease.

  1. TLRs: Perkins et al. [https://doi.org/10.1089/jir.2016.0003] reported a review of the status of the chromatin profile expressions of inflammatory genes in response to TLR stimulation and how this chromatin landscape affects cell type specific and temporal gene expression. Notably, TLR can induce epigenetic changes constituting both the positive and negative regulation of TLR‐induced genes. However, signaling pathways and transcription factors play a pivotal role in chromatin‐based mechanisms affecting proinflammatory responses and epigenetic marks in rewiring context‐specific gene expressions in innate immune cell types. Please expand.

The aspects that the reviewer states are very important. The manuscript provides a comprehensive description of the role that macrophages play on the disease, and the potential modulation of their phenotype by immunonutritional factors. The genetic basis of the macrophages’ differentiation would represent enough material to write an independent document. This basis would be dependent of, rather than the immunonutritional factor, the receptor activated. In this study we focused on the relationship between immunonutritional factors and the receptor that could be implicated.

  1. MetSyn: Please expand on the combined effects of diet and exercises on metabolic diseases through epigenetic modifications. What is the molecular mechanism involved?

Dear reviewer, as indicated previously, we could expect ‘epigenetic’ modifications in the macrophages. It is known that macrophages undergo (re)programming processes, which are responsible for the so called ‘trained immunity’ (Nat Rev Nephrol 19, 23–37 (2023). However, the exact mechanism(s) remains incomplete. In the current study, we focused on the macrophages’ activity and subphenotype as relevant determinants of the disease severity.

Reviewer 2 Report

Comments and Suggestions for Authors

The article entitled 'Revisiting the immunometabolic basis for the metabolic syndrome from an immunonutritional view' presents a thorough review of the metabolic syndrome (MetS), its immunometabolic basis and the potential role of immunonutritional factors (IFs). Here are the main shortcomings:

Missing research methodology:

The article lacks a clear section on research methodology. It is essential to include a detailed explanation of the methods used for the literature search, the criteria for selecting studies and the analytical approach for synthesising the results. This section is crucial for transparency and reproducibility.

Consideration of weight and abdominal fat:

Although the paper discusses various factors related to MetS, it does not consistently emphasise the importance of considering weight and abdominal fat. These are crucial components as they are directly linked to insulin resistance and cardiovascular risk, key aspects of the MetS.

Lack of practical implications and stimuli for new research:

The review does not provide clear practical implications for the findings or suggest directions for future research. It is important to highlight how the data reviewed may be applied in clinical practice or how they may shape future studies to advance understanding and treatment of MetS.

Confusing Sections:

Some sections of the document are difficult to follow due to scientific jargon and complex sentence structure. Simplifying these parts and using clear and concise language would improve readability. For example, sections discussing genetic predispositions and the interaction between genetic and environmental factors could be better structured to be clearer.

Excessive self-citation:

There is an over-reliance on self-citation, which can be perceived as biased. Although it is acceptable to cite one's own work, this should be balanced with references to other relevant studies in the field to provide a more complete and unbiased review.

To improve the article, authors should

Add a clear methodological section outlining the research process.

Emphasise the importance of weight and abdominal fat in the MetS discussion.

Include the practical implications of the findings and suggest future research avenues.

Simplify complex sections for better readability.

Reduce the number of self-citations and include a wider range of references.

Incorporating these suggestions would improve the clarity, credibility and applicability of the review.

Comments on the Quality of English Language

English is fine

Author Response

Reviewer #2

The article entitled 'Revisiting the immunometabolic basis for the metabolic syndrome from an immunonutritional view' presents a thorough review of the metabolic syndrome (MetS), its immunometabolic basis and the potential role of immunonutritional factors (IFs). Here are the main shortcomings:

Dear reviewer,

We really appreciate your efforts reviewing this study. In accordance with your comments, we enclose our answers. The pertinent changes to the manuscript have been made, and a list of the modifications is enclosed.

Missing research methodology:

The article lacks a clear section on research methodology. It is essential to include a detailed explanation of the methods used for the literature search, the criteria for selecting studies and the analytical approach for synthesising the results. This section is crucial for transparency and reproducibility.

We really appreciate your comment. Usually, description of the literature search is included only in systematic reviews, which is not the case of this document. According to the journal’s rules (https://www.mdpi.com/journal/biomedicines/instructions), which can be read at the webpage ‘Reviews offer a comprehensive analysis of the existing literature within a field of study, identifying current gaps or problems. They should be critical and constructive and provide recommendations for future research. No new, unpublished data should be presented. The structure can include an Abstract, Keywords, Introduction, Relevant Sections, Discussion, Conclusions, and Future Directions’.

As the reviewer can confirm it is not a need to include a section concerning research methodology. This is the reason why we did not include it.

According to the reviewer’s comment a brief sentence has been included in the manuscript (line 97-104).

Consideration of weight and abdominal fat:

Although the paper discusses various factors related to MetS, it does not consistently emphasise the importance of considering weight and abdominal fat. These are crucial components as they are directly linked to insulin resistance and cardiovascular risk, key aspects of the MetS.

We really appreciate your comment, which highlights a very important and indisputable factor contributing to the severity of the disease. However, >10% of lean individuals meet MetS criteria. Visceral adipose tissue, among others, it is recognized to disproportionately contribute to inflammation and insulin resistance (Front Immunol. 2021, 12:675018; Cells. 2022, 11(9):1435). Besides, it has also been reported that ‘Macrophage Phenotypes and Gene Expression Patterns Are Unique in Naturally Occurring Metabolically Healthy Obesity’ (Int J Mol Sci 2022, 23(20):12680). However, there is not any strategy targeting the activity of the monocytes-derived macrophages.

Previous research ruled out the assumption that visceral adipose tissue inflammation is increased in obese people with type 2 diabetes compared to high insulin resistance obese (PLoS One . 2012;7(10):e48155). Thus, abdominal fat is a very important factor but some additional processes appear to be responsible of other significant pathological consequences. In this study we approach the idea that innate immunity is a critical player and it can be modulated using food-based strategies, which today are not being considered.

In view of the reviewer’s concern it has been included a brief sentence to these respects in the introduction section (Line 75-80).

Lack of practical implications and stimuli for new research:

The review does not provide clear practical implications for the findings or suggest directions for future research. It is important to highlight how the data reviewed may be applied in clinical practice or how they may shape future studies to advance understanding and treatment of MetS.

Many thanks for your comment, which we really appreciate because it reveals the lack of immunonutritional approaches to treat/prevent the metabolic syndrome. A clear example of what we highlight is that checking the webpage of the ‘Mayo Clinic’ (https://www.mayoclinic.org/diseases-conditions/metabolic-syndrome/diagnosis-treatment/drc-20351921) it can be read (as treatment) ‘If aggressive lifestyle changes such as diet and exercise aren't enough, your doctor might suggest medications to help control your blood pressure, cholesterol and blood sugar levels.’

Currently, the strategies and approach stated in our manuscript it is poorly considered eban if the literature demonstrates the important consequences of controlling the immune system to reduce the severity of the disease. A quick search on PUBMED:GOV using simple words such as ‘immunity and metabolic syndrome’ shows 5,569 results. However, any single strategy focusing on how food can impact the activity of the innate branch of the immune to control the disease is being considered.

We consider these aspects a major contribution to the field and focusing on the molecular mechanisms a door is open to use them in the clinical practice. Notwithstanding, for us is clear, and completely agree with the reviewer that a significant bulk of data is needed to start using these strategies on human beings. From our side we already started these studies three years ago on our project, funded by the Spanish government, with Ref No: PID2019-107650RB-C22.

In view of the reviewer’s comment the last section approaching the ‘conclusions and future directions’ has been modified to clarify the potential clinical areas where immunonutritional strategies can contribute.

Confusing Sections:

Some sections of the document are difficult to follow due to scientific jargon and complex sentence structure. Simplifying these parts and using clear and concise language would improve readability. For example, sections discussing genetic predispositions and the interaction between genetic and environmental factors could be better structured to be clearer.

Taking into consideration the reviewer’s comment as well as those from the other reviewers, it has been introduced a figure trying to explain the interactions described in the manuscript (figure 1, line 175).

Excessive self-citation:

There is an over-reliance on self-citation, which can be perceived as biased. Although it is acceptable to cite one's own work, this should be balanced with references to other relevant studies in the field to provide a more complete and unbiased review.

We could agree as despite the importance of innate immune response(s) in the MetS there is an incomplete information about these on the scientific literature.

In the light of the reviewer’s comment some of our studies have been deleted.

To improve the article, authors should

  • Add a clear methodological section outlining the research process.
  • Emphasise the importance of weight and abdominal fat in the MetS discussion.
  • Include the practical implications of the findings and suggest future research avenues.
  • Simplify complex sections for better readability.
  • Reduce the number of self-citations and include a wider range of references.

Incorporating these suggestions would improve the clarity, credibility and applicability of the review.

Many thanks for your comments. The manuscript has been revised and modified accordingly to adjust it to these instructions.

Reviewer 3 Report

Comments and Suggestions for Authors

The article discusses the role of natural products as imunomodultors in metabolic syndrome. The review is very well structured and the themathic is very interesting.

Please discuss the role of imunmodulators that work as antioxidants and give some examples. Please check:

Alleviation of drugs and chemicals toxicity: Biomedical value of antioxidants, Oxid. Med. Cell. Longev., 2018, 2018. https://doi.org/10.1155/2018/6276438

 Please discuss what type of polyphenols are especially important in the cardiovascular complications of metabolic syndrome.

Please check:

Cardiovascular Risk and Statin Therapy Considerations in Women. Diagnostics 2020, 10(7), 483.

Comments on the Quality of English Language

Englissh is ok.

Author Response

The article discusses the role of natural products as imunomodultors in metabolic syndrome. The review is very well structured and the thematic is very interesting.

Dear reviewer,

We really appreciate your efforts reviewing this study. In accordance with your comments, we enclose our answers. The pertinent changes to the manuscript have been made, and a list of the modifications is enclosed.

Please discuss the role of imunmodulators that work as antioxidants and give some examples. Please check:

Alleviation of drugs and chemicals toxicity: Biomedical value of antioxidants, Oxid. Med. Cell. Longev., 2018, 2018. https://doi.org/10.1155/2018/6276438

Many thanks for your comment, which we really appreciate. The reason why antioxidants haven’t been included is because their activity results somehow ‘unspecific’ as they not exert the effects through a defined receptor. Besides, their bioavailability and distribution within the organism results quite variable. For example, SETIs are poorly absorbed but exert their activity via intestinal TLR4, which contributes to shape the gut microbiota composition and promoting an specific macrophages’ subphenotype.

In view of the reviewer’s comment the potential inmmunomodulatory role of the antioxidants has been briefly included in the document (line 76-86).

Please discuss what type of polyphenols are especially important in the cardiovascular complications of metabolic syndrome.

Please check:

Cardiovascular Risk and Statin Therapy Considerations in Women. Diagnostics 2020, 10(7), 483.

In consideration of the reviewer’s comment the importance of polyphenols influencing the macrophage’s polarization has been indicated in the introduction (line 76-86).

Round 2

Reviewer 1 Report

Comments and Suggestions for Authors

authors responded adequately to the reviewer's comments.

Comments on the Quality of English Language

minor English editing by Editorial staff.

Author Response

Dear reviewer, many thanks for your efforts and help to improve the quality and understanding of the manuscript

Reviewer 2 Report

Comments and Suggestions for Authors

Dear authors, if you do not wish to include the methodology, I suggest you at least to follow the SANRA guidelines for narrative reviews. This would make the paper more scientifically credible.

The number of self-citations must be a maximum of two.

Comments on the Quality of English Language

English is fine

Author Response

Dear authors, if you do not wish to include the methodology, I suggest you at least to follow the SANRA guidelines for narrative reviews. This would make the paper more scientifically credible.

Dear reviewer, we fully appreciate your comments. The reason why we did not include the methodology is, as said before, the journal’s rules. It is not that we do not want to indicate the methodology, but that the magazine does not indicate it that way.

According to the reviewer’s comment a new section describing the methodology used has been included in the manuscript.

We have revised our manuscript taking into consideration the SANRA-guidelines, and under our point of view the score is up to 9. This value is calculated assigning ‘0’ to the literature search (as the journal indicates)

  • The manuscript’s aim in stated in the line 104-107, now it has included a brief sentence indicating the lack of immunonutritional strategie lines 57-58.
  • As said, the journal’s web describes what they want. Now, this has been completed.
  •  
  • Key statements in the manuscript are properly references. For example, the critical role of the immune system determining diet-induced obesity and the role of macrophages on the energy expenditure are already described and referenced in lines 50-67.
  • A critical review of the literature reveals the lack of food-based strategies to selectively polarize the monocytes-derived macrophages and how they can influence insulin resistance. This is the reason why we used our studies as our major research line focuses on the idea to fill up this gap of knowledge. Now our studies have been taken out form the manuscript.

The number of self-citations must be a maximum of two.

According to the reviewers comment the self-citations have been adjusted to a maximum of 2.

Round 3

Reviewer 2 Report

Comments and Suggestions for Authors

The manuscript presents a narrative review on the topic of metabolic syndrome from an immunonutritional perspective. However, the methodology used for the review is non-existent or grossly inadequate. The SANRA (Scale for the Assessment of Narrative Review Articles) criteria were not followed, which severely compromises the quality and reliability of the work.

The main problems encountered are listed below:

Explicit Objectives: The objectives of the review are not clearly stated. There is no explicit formulation of the research questions that the manuscript intends to address.

Explicit Inclusion Criteria: No clear criteria are provided for the inclusion and exclusion of studies in the review. This leads to doubts about the selectivity and relevance of the reviewed literature.

Literature Collection and Selection: The methodology of literature collection and selection is neither systematic nor detailed. The description of the sources and search engines used is vague and lacks transparency.

Systematicity in Data Presentation: The data are not presented in a systematic and consistent manner. The review lacks a structured critical analysis of the included sources.

Critical Discussion:The discussion of the results is superficial and does not offer a thorough critique of the available evidence. Furthermore, the limitations of the included studies and potential sources of bias are not considered.

Relevant Conclusions: The conclusions are not well supported by the data presented and are not clearly aligned with the stated objectives (which in any case lack clarity).

Comments on the Quality of English Language

English is fine